# Design Space Exploration of Antenna Impedance and On-Chip Rectifier for Microwave Wireless Power Transfer

**Takuma Hashimoto and Toru Tanzawa \***

Graduate School of Integrated Science and Technology, Shizuoka University, Hamamatsu 432-8561, Japan
\* Correspondence: toru.tanzawa@shizuoka.ac.jp

**Abstract:** This paper discusses a design methodology to efficiently determine the best combination of rectenna (rectifier and antenna) to minimize the input power under a given output condition for microwave wireless power transfer (MWPT) without any other external components, such as a matching network, for cost reduction. A linearized equivalent circuit model is expanded upon to include the microstrip line connecting the antenna and rectifier. Based on the model, the design flow is presented that has mainly three steps: (1) Determination of the equivalent rectifier input impedance and the amplitude of input voltage by running SPICE simulation, (2) Drawing contour plots of input power by rectifier candidate on the antenna impedance plane by conducting model calculation and impedance loci of antenna candidates on the contour plots, and (3) Selecting the combination of antenna and rectifier which gives the minimum input power for all the combinations. To validate the equivalent circuit model and design flow, a single-diode (SD) rectifier and a voltage-doubler (VD) rectifier were fabricated in 65 nm CMOS. The input power to generate 100 µA at 1 Vdc was measured and compared. The model, SPICE and measurement are in good agreement with each other that VD has 30–50% lower input power than SD does. In addition, the sensitivity of the parasitic elements, such as the microstrip line and the bonding wires and pads on the input power, are investigated to explore the design space for rectenna.

**Keywords:** RF–DC converter; rectenna; microwave wireless power transfer; IoT





## 1. Introduction

In recent years, microwave wireless power transfer (MWPT) has attracted much attention as a method of powering sensor ICs of IoT devices [1]. The demand for IoT devices is predicted to increase explosively in the near future. If each of these devices is equipped with batteries, a large number of sensor ICs will require to be maintained for battery replacement, the cost of which can be significant. In addition, there are various applications where battery replacement itself is too difficult, such as medical devices implanted in the body [2]. MWPT can overcome these issues. The frequency of battery replacement can be reduced with MWPT. Ultimately, sensor ICs can be driven permanently without a battery. An MWPT system mainly consists of a transmitter for radiating electromagnetic (EM) waves and a receiver with a receiving antenna and a rectifier for capturing the radiated EM waves. This combination of receiving antenna and rectifier is called a rectenna [3]. Figure 1 shows a block diagram of a rectenna. In this research, it is considered that only the antenna is an external component and the other parts of the rectenna are integrated into a single sensor IC to minimize the cost, unlike a matching network that is used for impedance matching between the receiving antenna and the rectifier [4–6]. The antenna impedance is considered as one of the design parameters in this work. "Load" includes all the building blocks of the sensor IC except for the integrated rectifier.

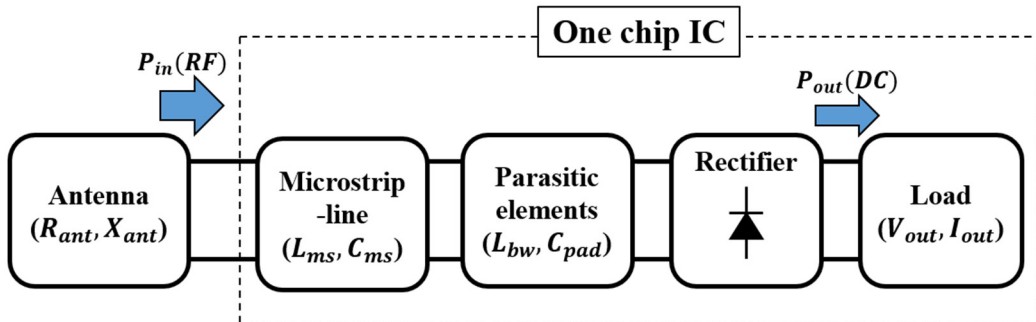

**Figure 1.** Block diagram of rectenna circuit for MWPT.

There have been several works to design rectenna in literature. In Ref. [7], a linearized model was proposed for Greinacher full-wave rectifier circuits with multiple stages. The proposed model can calculate the power conversion efficiency under a low power condition. In Ref. [4], an on-chip inductor was connected between the gate and drain of a diode-connected transistor to generate a large amplitude gate voltage by LC resonance with the gate parasitic capacitance. This composition is possible because of the EHF band. It would be very large in size if an on-chip inductor was used in the UHF band. In [8], a parasitic PN junction diode shorted by connecting bulk and drain was proposed to reduce the effective threshold voltage of a diode-connected transistor. This method increased the sensitivity of the rectifier. Refs. [4,8] focused on the integration of rectifiers, but did not discuss any design method to achieve high conversion efficiency including antennas. In Ref. [9], an optimization was conducted for a rectenna combining a square loop antenna and a cross-coupled rectifier. The sensitivity was shown to be −27 dBm to obtain 1 V across both terminals of a capacitive load. In Ref. [10], device parameters of rectifying diodes of RF–DC converters were discussed. As a result, to achieve higher conversion efficiency, diode junction capacitance is the most significant parameter at the micro-watt level. In Ref. [11], it was reported that a low input power rectenna was realized with serially connected diodes, which was an AC-DC charge pump without pumping capacitors for reducing the input capacitance. However, [10,11] assumed that the antenna impedance was given in advance, which does not indicate that such methods can provide a general optimum design for a rectenna. Depending on the application, the operation frequency is wide [12] or narrow [13].

A longer transmission distance is generally desired by MWPT users to have a wider service area. Increasing the transmission distance in MWPT can be realized by reducing the input power to the receiving antenna under the condition where the sensor IC can be driven with a specific average current at a specific voltage, as shown in Figure 2. In this research, the purpose is to effectively select the best combination that minimizes $P_{in}$ among combinations of multiple types of antennas and rectifiers without running many circuit simulations. The power required to drive the building blocks is given by the output voltage $V_{out}$ and the average output current $I_{out}$. Therefore, once a target IC is specified, the input power to the IC, i.e., the output power of rectifier $P_{out}$, can be determined. In addition, if the rectifier can be modeled with a linear circuit defined at $V_{out}$ and $I_{out}$, one will be able to have a relationship between $P_{in}$, $V_{out}$ and $I_{out}$ as a function of circuit parameters as shown in Figure 1. In Ref. [14], a linearized equivalent circuit model was proposed to effectively determine the best one among the combinations of multiple types of antennas and rectifiers to have the minimum input power.

This paper expands upon the linearized equivalent circuit model to include the microstrip line connecting the antenna and the rectifier and the parasitic resistance of the silicon substrate. In addition, the efficient design flow using the models is also shown in more detail. Experimental results are also presented to validate the model. Furthermore, the impact of the parasitic elements of the microstrip line and the bonding wires and pads on the input power is investigated to explore the design space for the rectenna. The main

contribution of the proposed method is improving the efficiency of rectenna design. For example, when one tries to run circuit simulations for N sizes of each of N types of antenna connected with N types of rectifiers, a straightforward design flow requires $N^3$ simulation runs, whereas a proposed design flow carries out $N^1$ simulation runs only for N types of rectifiers. The best combination can be determined with $N^3$ model calculation.

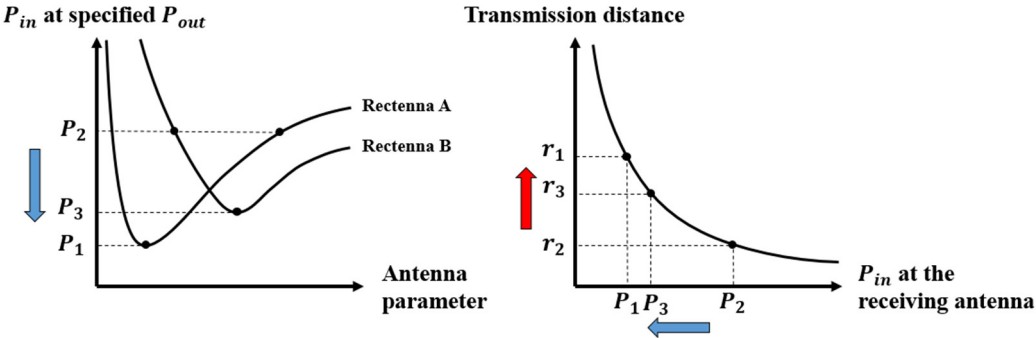

**Figure 2.** Antenna parameters vs. $P_{in}$ and $P_{in}$ vs. transmission distance.

This paper is organized as follows. Section 2 proposes a co-design flow for the antenna and on-chip rectifier and describes circuit linearization and formulation. Section 3 conducts a demonstration of the proposed design flow. Section 4 compares the results of model calculation, SPICE simulation and measurement to validate the performed demonstration in Section 3. Section 5 discusses the influence of the parasitic components on the input power. Section 6 summarizes this research.

## 2. Antenna/On-Chip Rectifier Optimum Co-Design Flow

In this section, linearized equivalent circuit models are discussed to have an equation relating $P_{in}$ with $V_{out}$ and $I_{out}$ as a function of the circuit parameters of all the components. This equation will realize a design flow in the backward direction to design the rectenna effectively. Table 1 summarizes the circuit parameters of the rectenna.

**Table 1.** Description of parameters.

| Parameter | Description | Parameter | Description |
|---|---|---|---|
| $R_R$ | Equivalent input resistance of a rectifier | $C_{pad}$ | Parasitic capacitance of a pad |
| $C_R$ | Equivalent input capacitance of a rectifier | $L_{ms}$ | Parasitic inductance of a microstrip line |
| $V_{Rec}^{pk}$ | Peak input voltage of a rectifier | $C_{ms}$ | Parasitic capacitance of a microstrip line |
| $R_{ant}$ | Real part of the antenna impedance | $L_{bw}$ | Parasitic inductance of a bonding wire |
| $X_{ant}$ | Imaginary part of the antenna impedance | $C$ | Capacitance of each capacitor |
|  |  | $N$ | Number of capacitors |

Figure 3 shows a straightforward design flow for rectenna using a circuit simulator, which is called "design flow in the forward direction" in this paper. The basic idea is as follows:

In Step 1, an antenna is assumed to have $R_{ant}$ and $X_{ant}$ within predetermined ranges. Similarly, a rectifier is assumed to have $C$ and $N$ within predetermined ranges.

In Step 2, the rectenna designer runs a circuit simulator by varying the input power to output the voltage at a specific load until the input power required to generate a specific voltage at the load is determined.

In Step 3, the data set of the antenna ($R_{ant}$ and $X_{ant}$), the rectifier ($C$, $N$) and $P_{in}$ is recorded. Then, another combination of antenna and rectifier is assumed.

Steps 1–3 are repeated until all the combinations are simulated. Among the values of $P_{in}$, the lowest one is identified. At the same time, one can determine the best combination of antenna and rectifier. As a result, the rectenna designer needs to run the simulator as much as the number of combinations of antennas and rectifiers. For example, if the

rectenna designers want to investigate N-antennas and N-rectifiers, they need to run $N^2$-simulations. Therefore, this "design flow in the forward direction", i.e., "$P_{in}$ input $P_{out}$ output" is very time-consuming.

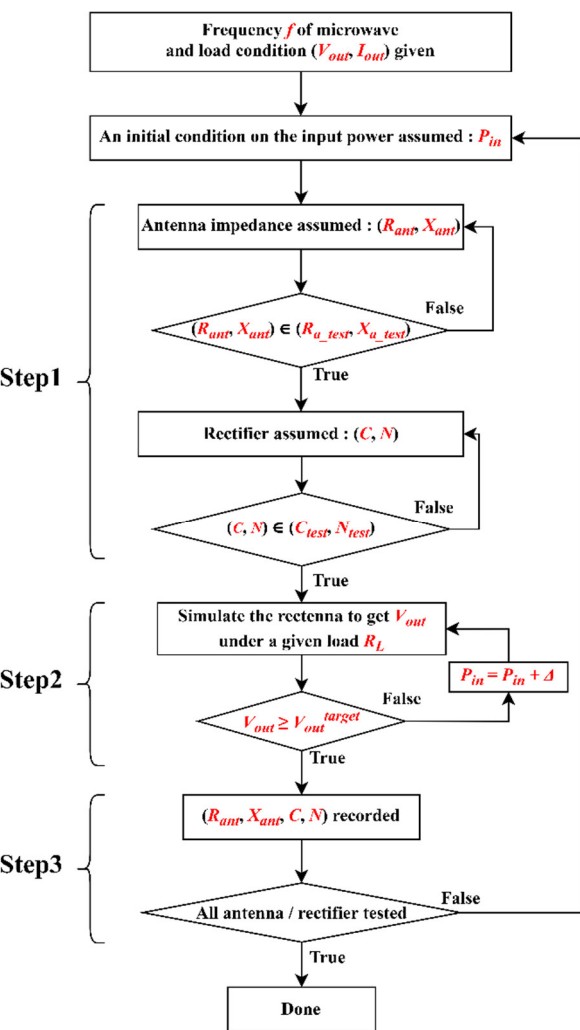

**Figure 3.** A design flow with multiple feedback loops in the forward direction.

Thus, "design flow in the backward direction" was proposed [14], as shown in Figure 4. In the backward direction flow, the input power is calculated by using a linearized model with the output power given in advance. The proposed design flow has mainly three steps: (Step 1) determination of $R_R$ and $V_{Rec}^{pk}$ for each rectifier candidate, (Step 2) drawing the contour plots of input power by conducting model calculation for each rectifier candidate on antenna impedance plane ($R_{ant} - X_{ant}$) and impedance loci of the antenna candidates on the contour plots, and (Step 3) exploring the condition that gives the minimum input power among all the combinations by comparing the minimum input power of each combination, considered in Step 2. Thus, circuit simulations are needed only for rectifiers. As a result, even if the rectenna designers want to investigate N-antennas and N-rectifiers, they only need to run N-simulations. $N^2$-combinations are investigated with model calculations which are much less time-consuming. Therefore, this "design flow in the backward direction", i.e., "$P_{out}$ input $P_{in}$ output" is very effective in respect of time. For example, when one tries to run circuit simulations for N sizes of each of N types of antenna connected with N types of rectifiers, a straightforward design flow requires $N^3$ simulation runs whereas a proposed design flow does $N^1$ simulation runs only for N types of rectifiers. The best combination can be determined with $N^3$ model calculation. This means that the

proposed design flow increases the design efficiency by $N^2$, assuming the cost for $N^3$ model calculation is negligibly small.

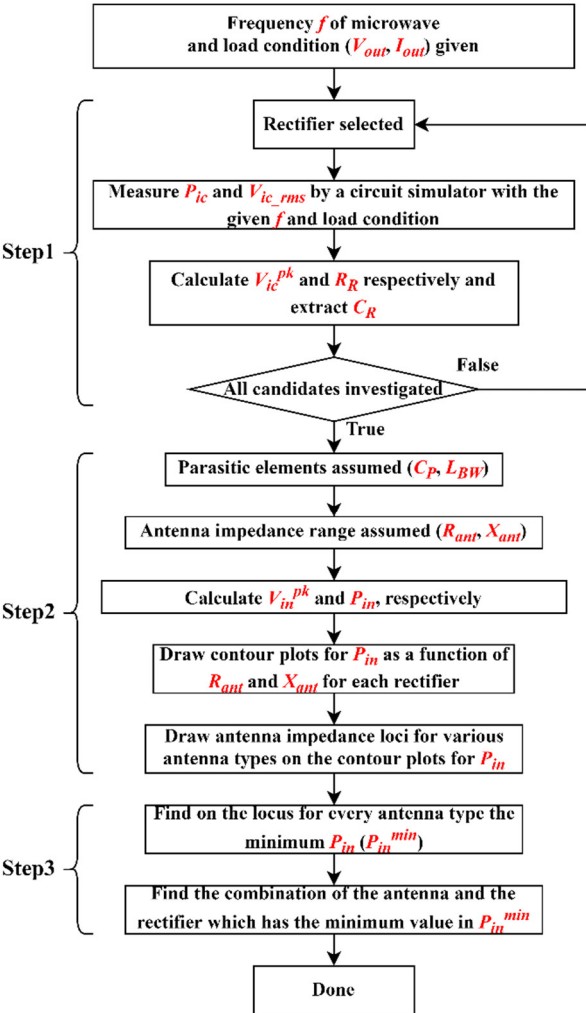

**Figure 4.** A proposed design flow for simultaneous optimization with a single feedback loop in the backward direction.

Each step will be explained in more detail as follows.

### 2.1. (Step 1) Determination of $R_R$ and $V_{Rec}^{pk}$

First, $V_{out}$ and $I_{out}$ specified by a sensor IC are given in advance. Figure 5a shows a rectifier and load that models the sensor IC. The input voltage amplitude of the rectifier $V_{Rec}^{pk}$ can be calculated from Equation (1). Here, $N$ and $V_{th}$ represent the number of stages of the rectifier and the threshold voltage of transistors used as rectifying diodes, respectively.

$$V_{Rec}^{pk} = \frac{V_{out}}{N} + V_{th} \tag{1}$$

The input impedance of the rectifier $Z_R$ can be determined by running a circuit simulator such as SPICE. An equivalent circuit model for the rectifier followed by the load can be expressed by a parallel circuit of $R_R$ and $C_R$, as shown in Figure 5b, because the imaginary part of the input impedance of Figure 5a is negative. The values of $R_R$ and $C_R$ can be calculated from the waveforms of the input voltage $V_{Rec}$, the input current $I_{Rec}$ and the power into the rectifier $P_{Rec}$ when $I_{out}$ is as high as the specified value at the target $V_{out}$ as follows. Figure 6 shows the waveforms simulated with SPICE. The rectenna circuit was

composed of voltage doubler (VD) described later as a rectifier, ESD protection diodes, $C_{pad}$ of 0.3 pF and $L_{bw}$ of 8 nH at 920 MHz, $I_{out} = 100$ μA and $V_{out} = 1$ V. Blue and orange dashed lines mean peak voltage of $V_{Rec}$ and average power of $P_{Rec}$, respectively.

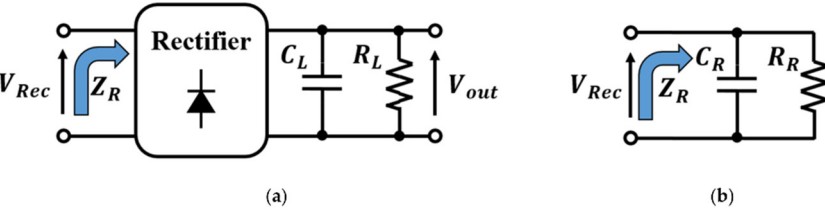

(a)    (b)

**Figure 5.** (**a**) Circuit of rectifier and load and (**b**) linearized model.

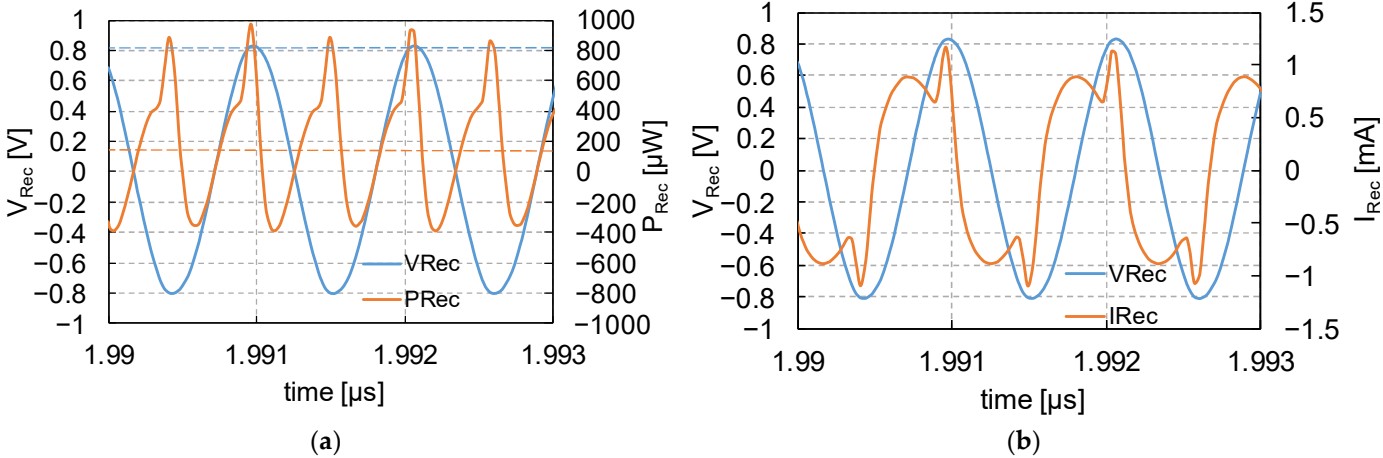

(**a**)    (**b**)

**Figure 6.** Waveforms of $V_{Rec}$ and $P_{Rec}$ (**a**) and of $V_{Rec}$ and $I_{Rec}$ (**b**).

A value of $R_R$ is determined by the effective voltage $V_{Rec-RMS}$ and the average power $\overline{P_{Rec}}$, as shown in Equation (2).

$$R_R = \frac{V_{Rec-RMS}^2}{\overline{P_{Rec}}} \tag{2}$$

$I_{Rec}$ includes the current flowing through diodes and the AC current through the capacitors. A value of $C_R$ is determined by the phase difference between $V_{Rec}$ and the AC current of $I_{Rec}$. The input impedance of a parallel RC model circuit, as shown in Figure 5b, is given by Equation (3).

$$\begin{aligned} Z_R &= \frac{R_R}{1+j\omega C_R R_R} \\ &= \frac{R_R}{1+(\omega C_R R_R)^2} - j\frac{\omega C_R R_R^2}{1+(\omega C_R R_R)^2} \end{aligned} \tag{3}$$

The phase difference between voltage and current $\phi$ is associated with $R_R$ and $C_R$.

$$\begin{aligned} \phi &= \arctan\left(\frac{Im(Z_R)}{Re(Z_R)}\right) \\ &= -\omega C_R R_R \end{aligned} \tag{4}$$

$C_R$ can be uniquely determined using $\phi$ obtained by the SPICE waveforms and $R_R$ obtained from (2).

### 2.2. (Step 2) Drawing Contour Plots of Input Power and Antenna Impedance Loci

In this paper, two parasitic models are considered. Model 1 includes bonding wires and bonding pads as parasitic components, as shown in Figure 7a. $C_{pad}$ includes $C_R$ and the parasitic capacitance of ESD protection circuit [9,15,16]. Model 2 further includes the

microstrip line expressed by $L_{ms}$ and $C_{ms}$ and the substrate resistance connected with the pad capacitance in series, as shown in Figure 7b. $C_{esdp}$ includes $C_R$ and the parasitic capacitance of ESD protection circuit.

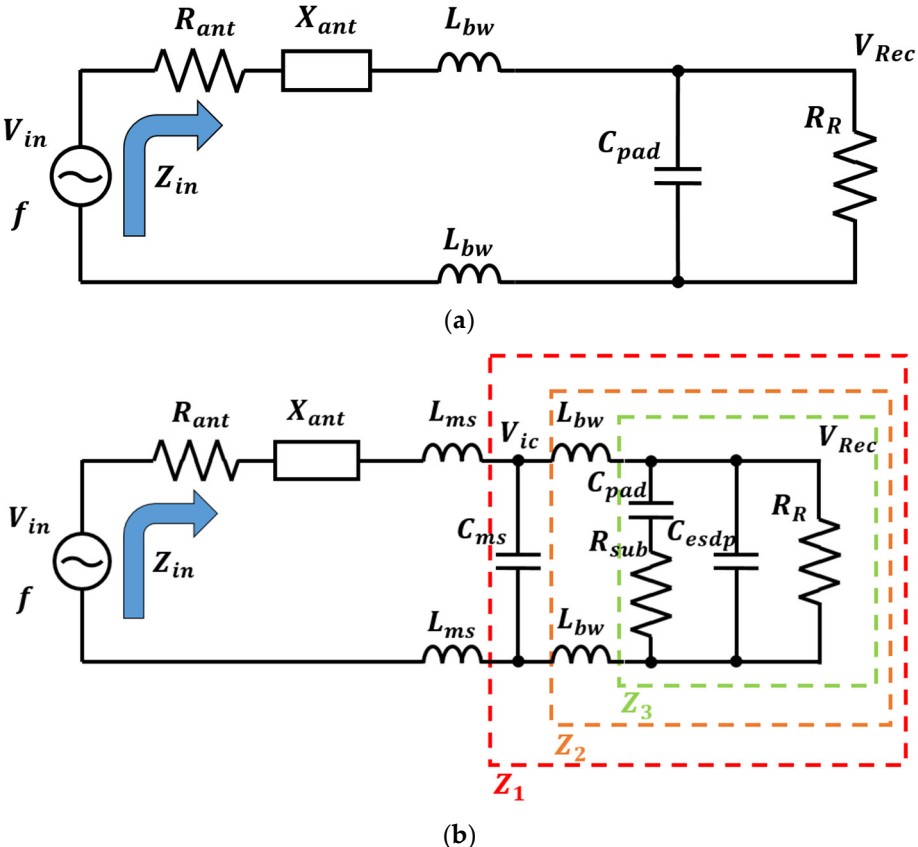

**Figure 7.** (**a**) Simple parasitic model (Model 1), (**b**) Detailed parasitic model (Model 2).

The input power $P_{in}$ is derived as follows.

(a)   Pin with Model 1

The input voltage amplitude at the antenna $V_{in}^{pk}$ is expressed by Equation (5) based on the model circuit of Figure 7a, where $V_{Rec}^{pk}$ is determined by Equation (1) and $X_a$ represents $X_{ant} + 2\omega L_{bw}$.

$$V_{in}^{pk} = \frac{V_{Rec}^{pk}}{R_R} \sqrt{\left( R_{ant} + R_R - \omega X_a C_{pad} R_R \right)^2 + \left( X_a + \omega C_{pad} R_{ant} R_R \right)^2} \tag{5}$$

The apparent input power $P_{in-app}$ can be given by Equation (6).

$$P_{in-app} = \frac{V_{in}^{pk}}{2} \frac{1}{\sqrt{\left( R_{ant} + \frac{R_R}{1+\left(\omega C_{pad} R_R\right)^2} \right)^2 + \left( X_a - \frac{\omega C_{pad} R_R^2}{1+\left(\omega C_{pad} R_R\right)^2} \right)^2}} \tag{6}$$

The effective input power $P_{in-eff}$ can be given by Equation (8).

$$\theta = \arctan(\arg(Z_{in})) \tag{7}$$

$$P_{in-eff} = P_{in-app} \cos \theta \tag{8}$$

(b)   Pin with Model 2

Using $Z_{in}$, $Z_1$, $Z_2$ and $Z_3$ as shown in Figure 7b, the amplitude of the intermediate voltage $V_{ic}^{pk}$ and $V_{Rec}^{pk}$ can be expressed by Equations (9) and (10), respectively.

$$V_{ic}^{pk} = \frac{|Z_1|}{|Z_{in}|} V_{in}^{pk} \tag{9}$$

$$V_{Rec}^{pk} = \frac{|Z_3|}{|Z_2|} V_{ic}^{pk} \tag{10}$$

From (9) and (10),

$$V_{in}^{pk} = \frac{|Z_{in}||Z_2|}{|Z_1||Z_3|} V_{Rec}^{pk} \tag{11}$$

$Z_{in}$, $Z_1$, $Z_2$ and $Z_3$ are expressed as follows, where $X_b$ represents $X_{ant} + 2\omega L_{ms}$.

$$Z_{in} = R_{ant} + jX_b + Z_1 \tag{12}$$

$$Z_1 = \frac{Z_2}{1 + j\omega C_{ms} Z_2} \tag{13}$$

$$Z_2 = j\omega L_{bw} + Z_3 \tag{14}$$

$$Z_3 = \frac{R_R + j\omega C_{pad} R_{sub} R_R}{\left(1 - \omega^2 C_{pad} C_{esdp} R_{sub} R_R\right) + j\left(\omega C_{pad}(R_R + R_{sub}) + \omega C_{esdp} R_R\right)} \tag{15}$$

$P_{in-app}$ is given by (16).

$$P_{in-app} = \frac{V_{in}^{pk2}}{2|Z_{in}|} \tag{16}$$

One can calculate $P_{in-app}$ using Equation (1), $C_R$, $R_R$ values determined in Step 1, and Equations (11)–(16) when $V_{out}$ and $I_{out}$ are specified. $P_{in-eff}$ can be obtained from Equations (7), (8) and (16).

The input power contour plots can be drawn on $R_{ant} - X_{ant}$ plane for each of model 1 and 2 using Equations (6), (8) and (16). The antenna impedance is determined by the shape of the antenna and its size. By varying the size of an antenna, the impedance loci of the antenna can be drawn over the input power contour plots. One can draw multiple impedance loci of various antenna shapes, as shown in Figure 8.

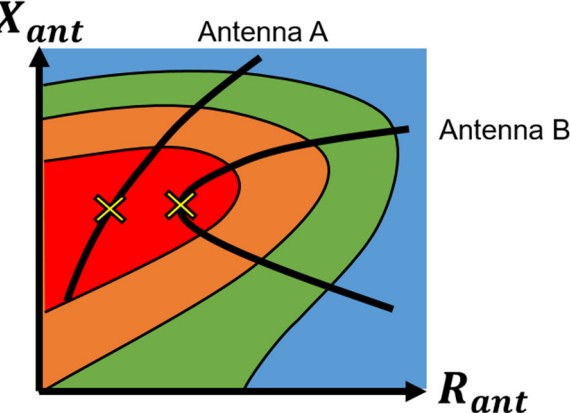

**Figure 8.** Image of the optimum design flow in steps 2 and 3.

### 2.3. (Step 3) Exploration of Combination of Antenna and Rectifier That Gives Minimum Input Power

One can determine the minimum input power among the least power by antenna shape expressed by "X" in Figure 8. Such data can be collected for various rectifiers with a different number of stages and different capacitance values of the capacitors. Among the

combinations of antenna and rectifier, one can identify the best one to have the minimum input power. Then, precise design can be conducted by running a circuit simulation around the condition identified by the design flow.

Assuming that one had N different types of rectifiers and N different types of antennas, in order to identify the best combination to have the minimum input power under the condition of given $V_{out}$ and $I_{out}$, one needed to run $N^2$ circuit simulations without the proposed design flow. With the proposed design flow, one should need to run N circuit simulations for N different types of rectifiers to determine $R_R$ and $V_{ic}^{pk}$ by rectifier. One can determine the best combination with $N^2$ calculations based on the linearized model. Thus, an initial circuit to be optimized can be determined highly efficiently.

### 3. Demonstration

In this section, a demonstration for the proposed design flow is shown. Two types of antennas, dipole antenna and circular loop antenna, as shown in Figure 9, and two types of rectifiers, single diode rectifier (SD) and voltage doubler (VD) are considered (Figure 10).

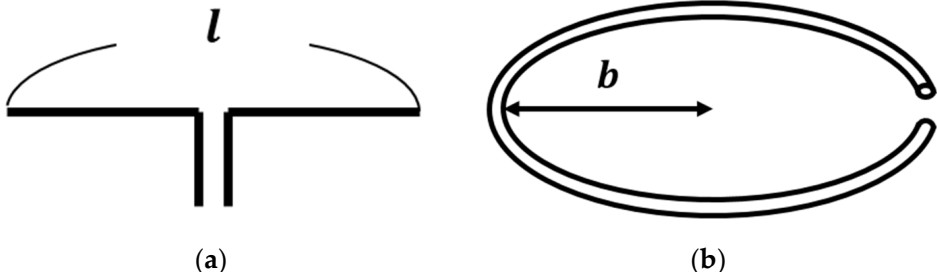

(a)                                                    (b)

**Figure 9.** Antenna candidates: (**a**) dipole antenna and (**b**) circular loop antenna.

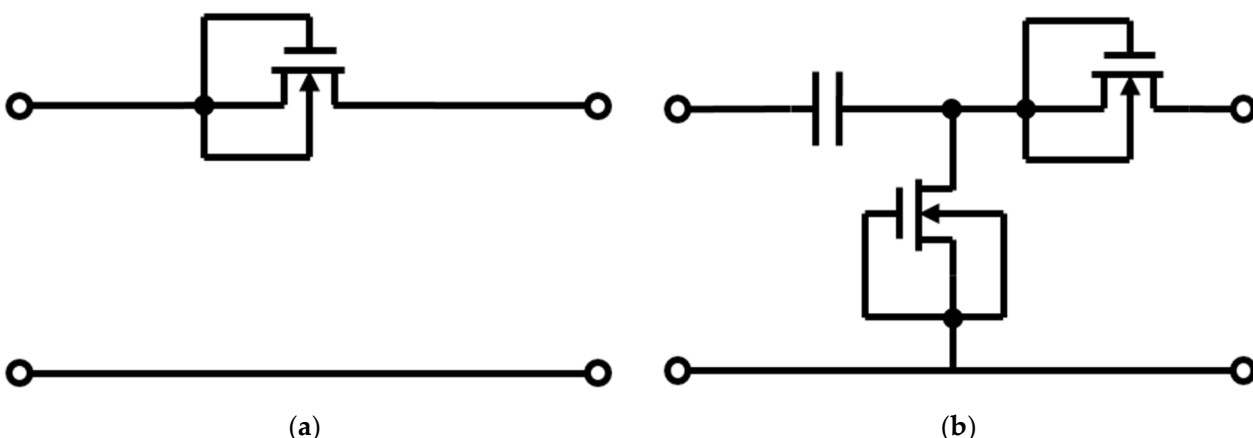

(a)                                                    (b)

**Figure 10.** Rectifier candidates: (**a**) single diode rectifier (SD) and (**b**) voltage doubler (VD).

The load is assumed to be 10 kΩ where the sensor/RF blocks require $I_{out}$ of 100 μA at $V_{out}$ of 1 V. It is also assumed that an input frequency of 920 MHz is used for MWPT. Parameters of the diode-connected MOSFETs, input capacitance of VD and parameters of parasitic components are summarized in Table 2, where 65 nm CMOS technology is assumed. The input capacitance of VD is used to charge the load when the input voltage is positive and boost the DC voltage when the input voltage is negative.

$R_R$ and $C_R$ can be determined by running SPICE for the sub-circuits with SD and VD, as shown in Figure 5a. $V_{Rec}^{pk}$ can be calculated by Equation (1) for each rectifier candidate. The results are summarized in Table 3.

Figure 11 shows the contour plots with Model 1 and SPICE simulation when the rectifier type is VD. Because it takes enormous time to run SPICE in small stride for a vast antenna impedance region, we analyzed the number of combinations of $R_{ant}$

and $X_{ant}$ and limited it to 27, i.e., $R_{ant} = 1 \sim 100\,\Omega$, and $X_{ant} = -100 \sim 100\,\Omega$ at $P_{out} = 100\,\mu\text{W}$ (1 V, 100 μA) for each parasitic circuit model in the following graphs. The model calculations were carried out using Equations (6), (8) and (16) derived in Section 2. Similarly, Figure 12 shows the ones with Model 2.

**Table 2.** Various parameters used in the demonstration.

| Parameters | Value |
|---|---|
| Threshold voltage of transistor $V_{th}$[V] | 0.4 |
| Input capacitance of VD $C_{in}$[pF] | 5.0 |
| $L_{ms}$[nH] | 4.3 |
| $C_{ms}$[pF] | 0.5 |
| $L_{bw}$[nH] | 8.0 |
| $C_{pad}$[pF] | 1.3 |
| $R_{sub}$[Ω] | 6.0 |
| $C_{esdp}$[pF] | 0.2 |

**Table 3.** $V_{Rec}^{pk}$, $R_R$, $C_R$ for each rectifier candidate at $I_{out} = 100\,\mu\text{A}$ and $V_{out} = 1\,\text{V}$.

| Rectifier Type | $V_{Rec}^{pk}$[V] | $R_R$[Ω] | $C_R$[fF] |
|---|---|---|---|
| SD ($N = 1$) | 1.4 | 5040 | 31.2 |
| VD ($N = 2$) | 0.9 | 2060 | 114 |

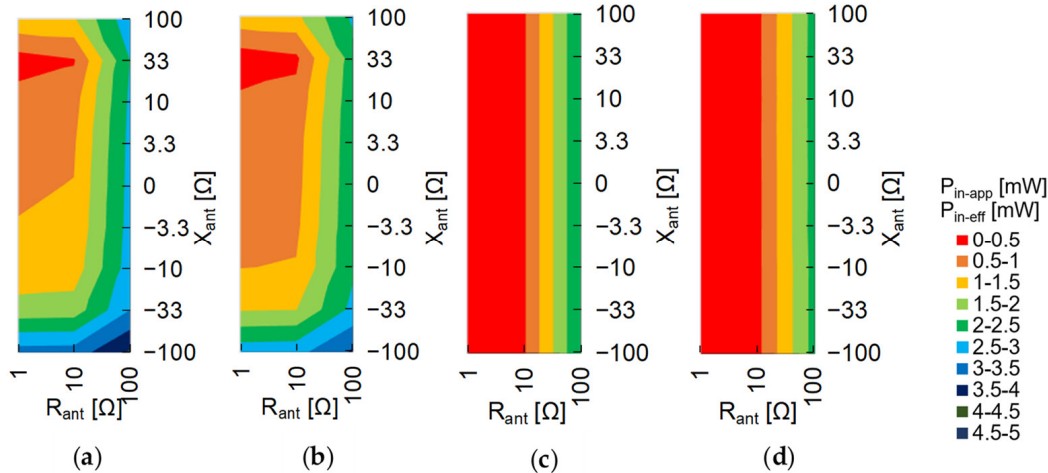

**Figure 11.** Contour plots of the input power in case of Model 1 with limited impedance points: (**a**) model calculation of $P_{in-app}$, (**b**) SPICE simulation of $P_{in-app}$, (**c**) model calculation of $P_{in-eff}$, (**d**) SPICE simulation of $P_{in-eff}$.

The trends shown by the models are well matched with the SPICE results. Therefore, the proposed design equations are proved to be reliable to calculate $P_{in}$ on the impedance plane. The antenna impedance varies with the element length for dipole antenna and the loop diameter for circular loop antenna [14]. Therefore, their antenna impedance loci can be drawn on the input power contour plots ($P_{in-app}$ and $P_{in-eff}$) of each rectifier candidate, as shown in Figures 13 and 14 for models 1 and 2, respectively. The model calculations were conducted for a wider range of $R_{ant} = 1 \sim 1000\,\Omega$ and $X_{ant} = -1000 \sim 1000\,\Omega$ for SD and VD, respectively, because the model calculations do not take much more time than the circuit simulator does. The impedance loci of dipole antenna and circular loop antenna were also plotted on the contour plots for each rectifier candidate.

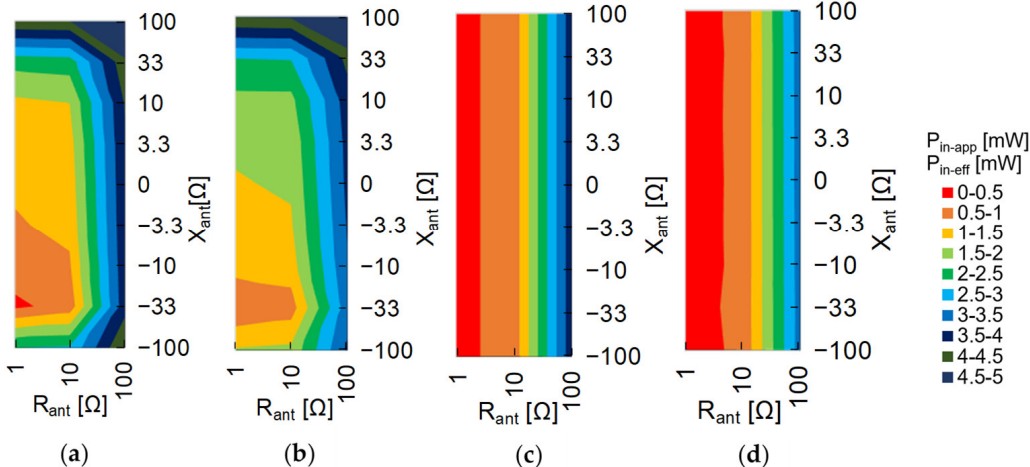

**Figure 12.** Contour plots of the input power in case of Model 2 with limited impedance points: (**a**) model calculation of $P_{in-app}$, (**b**) SPICE simulation of $P_{in-app}$, (**c**) model calculation of $P_{in-eff}$, (**d**) SPICE simulation of $P_{in-eff}$.

**Figure 13.** Contour plots of input power with model 1 at $P_{out} = 100$ μW: (**a**) $P_{in-app}$ (SD), (**b**) $P_{in-eff}$ (SD), (**c**) $P_{in-app}$ (VD), (**d**) $P_{in-eff}$ (VD).

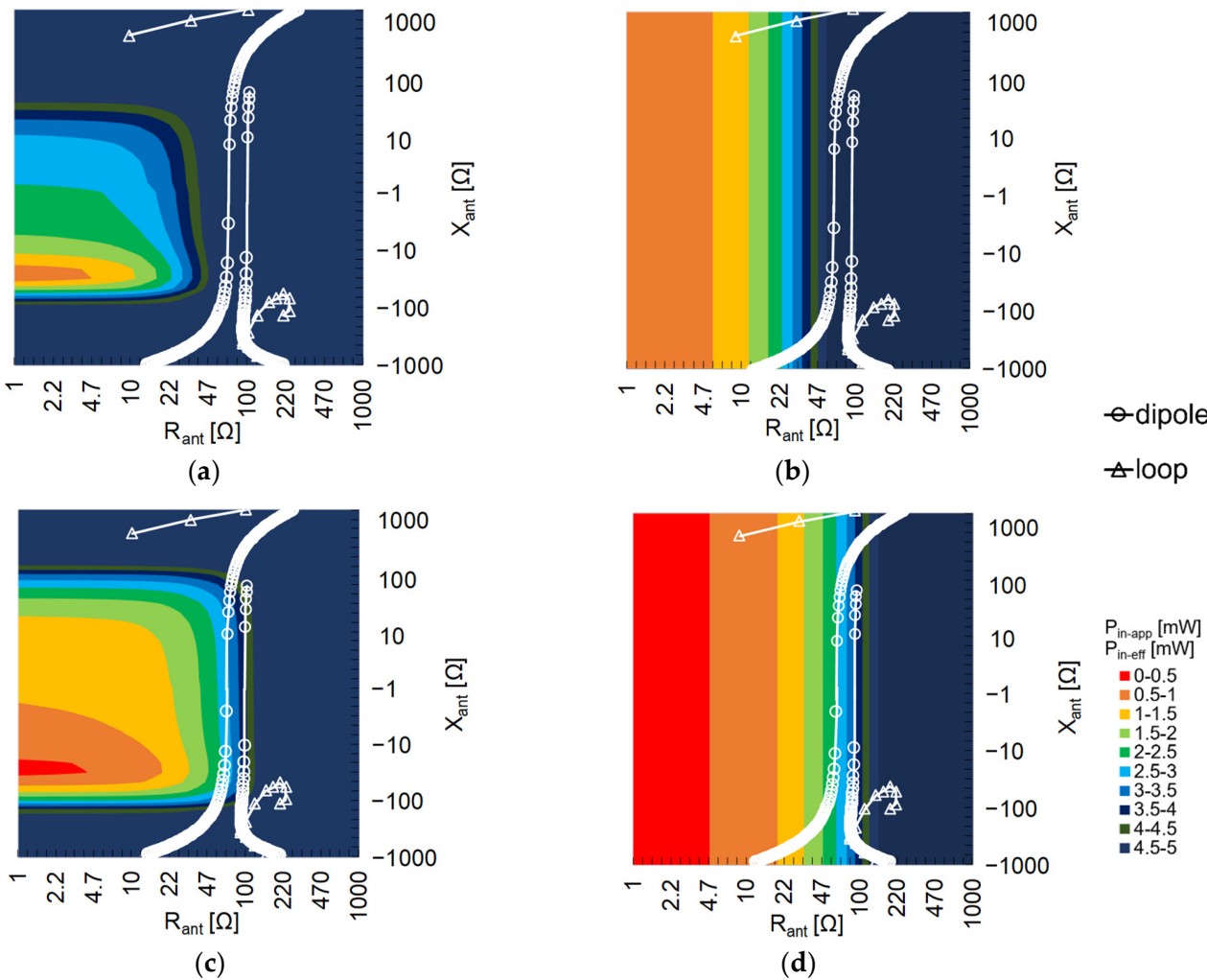

**Figure 14.** Contour plots of input power with Model 2 at $P_{out} = 100$ µW: (**a**) $P_{in-app}$ (SD), (**b**) $P_{in-eff}$ (SD), (**c**) $P_{in-app}$ (VD), (**d**) $P_{in-eff}$ (VD).

Let us focus on the apparent power because it represents the power that needs to be input into the antenna. The antenna impedance $Z_{ant\_min}$ to minimize the input power for each rectifier and the lowest input power $P^{match}_{in-eff}$ are summarized in Table 4.

**Table 4.** Antenna impedance to minimize the input apparent power $P^{match}_{in}$ and the minimum input power for each rectifier.

| Rectifier | (Model 1) | | (Model 2) | |
|---|---|---|---|---|
| | $Z_{ant\_min}[\Omega]$ | $P^{match}_{in-eff}[mW]$ | $Z_{ant\_min}[\Omega]$ | $P^{match}_{in-eff}[mW]$ |
| SD | 1 + j47 | 0.250 | 1 − j33 | 0.609 |
| VD | 1 + j47 | 0.219 | 1 − j33 | 0.368 |

If one can design the rectenna whose impedance is as much as $Z_{ant\_min}$, one can have the least input power as shown by $P^{match}_{in-eff}$ in Table 4. In this case, the highest power conversion efficiency defined by $P_{out}/P^{match}_{in-eff}$ is as high as 46%. However, in the demonstration, both dipole and loop antennas cannot have $Z_{ant\_min}$, much higher input power is required as shown in Table 5, where $l$, $b$ and $\lambda$ represent element length, loop radius and wavelength of EM waves, respectively. Table 5 indicates that a combination of VD and dipole antenna has the minimum input power regardless of the model. Model 1 needs $P^{min}_{in} = 1.85$ mW at $l/\lambda = 0.497$ ($l = 16.2$ cm, $\lambda = 32.6$ cm) and Model 2 needs

$P_{in}^{min} = 2.57$ mW at $l/\lambda = 0.478$ ($l = 15.6$ cm, $\lambda = 32.6$ cm). These results were obtained without multiple circuit simulations by trial and error. To confirm the validity of models 1 and 2, SPICE simulations were run as well with various sizes of the antennas, as shown in Figures 15 and 16, respectively.

**Table 5.** Minimum input apparent power $P_{in}^{min}$ and antenna parameter at that condition for each combination.

| | | $P_{in}^{min}$[mW] | | | |
|---|---|---|---|---|---|
| | | (Model 1) | | (Model 2) | |
| **Rectifier** | | **Dipole($l/\lambda$)** | **Loop($2\pi b/\lambda$)** | **Dipole($l/\lambda$)** | **Loop($2\pi b/\lambda$)** |
| SD | | 4.20 (0.497) | 19.9 (0.34) | 5.95 (0.478) | 26.2 (0.8) |
| VD | | 1.85 (0.497) | 8.28 (0.34) | 2.57 (0.478) | 10.9 (0.8) |

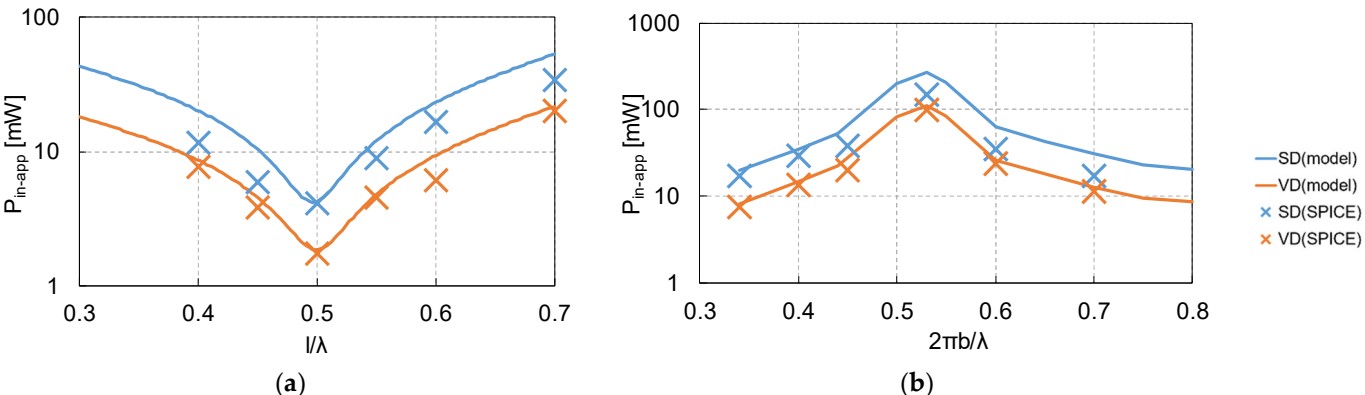

**Figure 15.** Antenna parameter vs. $P_{in-app}$ based on model 1 at $P_{out} = 100$ μW(1 V, 100 μA): (**a**) dipole antenna, (**b**) circular loop antenna.

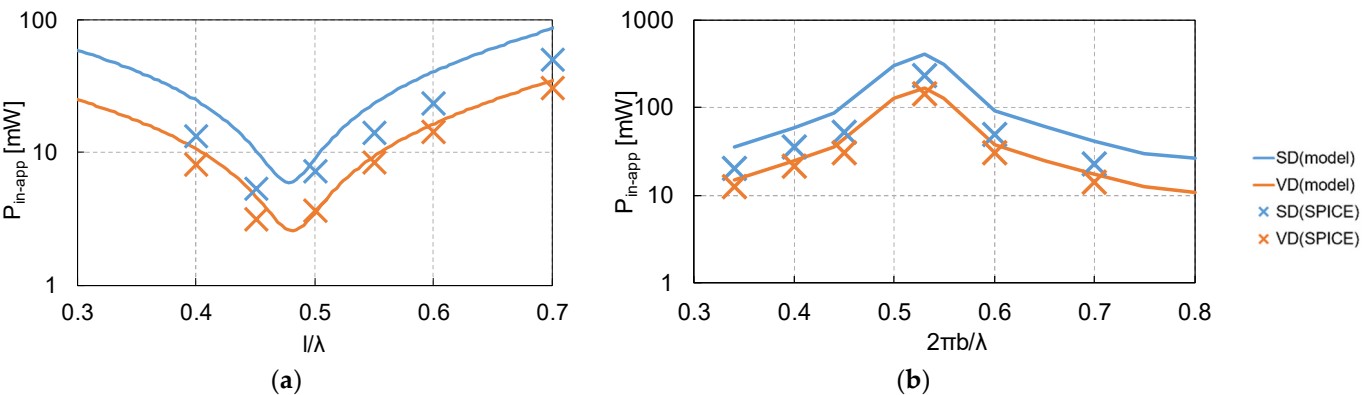

**Figure 16.** Antenna parameter vs. $P_{in-app}$ based on model 2 at $P_{out} = 100$ μW(1 V, 100 μA): (**a**) dipole antenna, (**b**) circular loop antenna.

The trends of the minimum input power across the size of the antennas with the models are in good agreement with SPICE results. Therefore, the models are useful to determine the size of the antenna which has the minimum input power. Note that the proposed design methodology in this research doesn't consider the directivity, gain and radiating efficiency of antennas. Hence, it is necessary to conduct an analysis that incorporates these factors. These are considerations for future works.

## 4. Verification of the Proposed Design Flow with Measurement

Two types of rectifiers (SD and VD) were fabricated in 65 nm CMOS to validate the models and design flow, as shown in Figure 17. Evaluation circuit boards had SMA

connectors to input power and microstrip lines between the SMA connectors and the packaged ICs. Therefore, Model 2 was used to compare the model calculation with the SPICE simulation. The input impedance of the evaluation board was measured using a vector network analyzer (VNA) to extract the parasitic parameters at 920 MHz, which are summarized in Table 2. A Smith chart of the input impedance at $0.5 \sim 1.5$ GHz is shown in Figure 18. The circle in red represents the input impedance at 920 MHz.

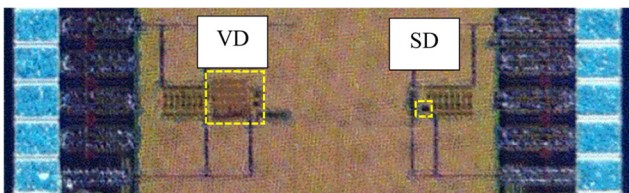

**Figure 17.** Die photo of SD and VD.

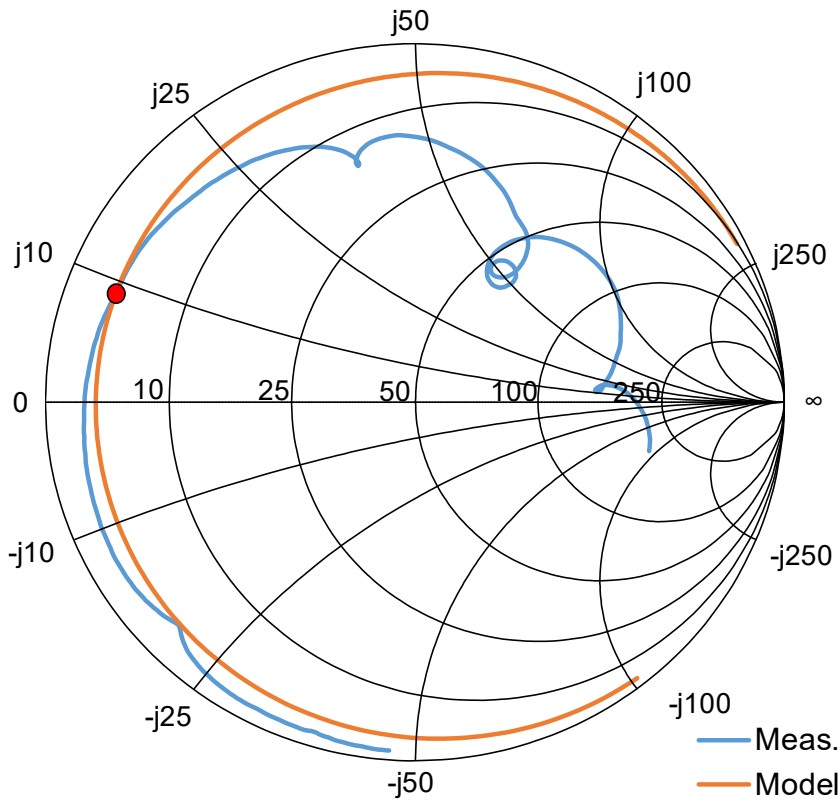

**Figure 18.** Input impedance of the evaluation board at the input frequency of $0.5 \sim 1.5$ GHz.

The silicon area was approximately 5200 μm$^2$ for VD and 150 μm$^2$ for SD, respectively. The difference was mainly from the input capacitance of VD, which is dominant.

Figure 19 shows a measurement setup. $P_{ref}$ and $P_{RF}$ represent the reflection power and output power from the RF generator, respectively. To estimate actual $P_{in-eff}$, the reflected power at the input port of the rectifier and the losses in connectors and cables were needed. Figure 20 shows the extraction method for the reflected power at the input port of the rectifier.

The losses in connectors and cables were measured using an RF generator and spectrum analyzer. Thus, $P_{in-eff}$ was estimated. To estimate $P_{in-eff}$, the power factor $cos\theta$ was needed. The input impedance of the rectifier was measured by VNA. As a result, $P_{in-eff}$ under 50 Ω system was estimated as shown in Figure 21. $P_{in-app}$ for VD was lower by 56.8% (model calculation), 37.1% (SPICE) and 49.3% (measurement) than that for SD.

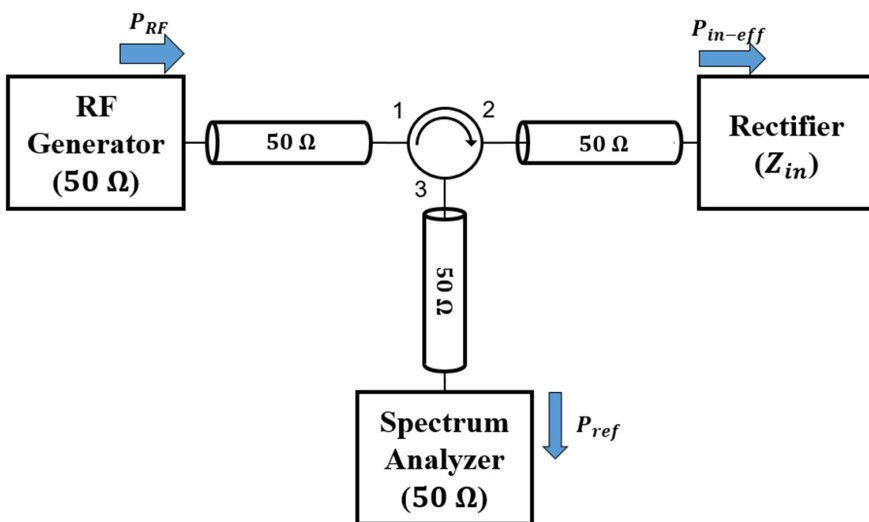

**Figure 19.** Block diagram of measurement system.

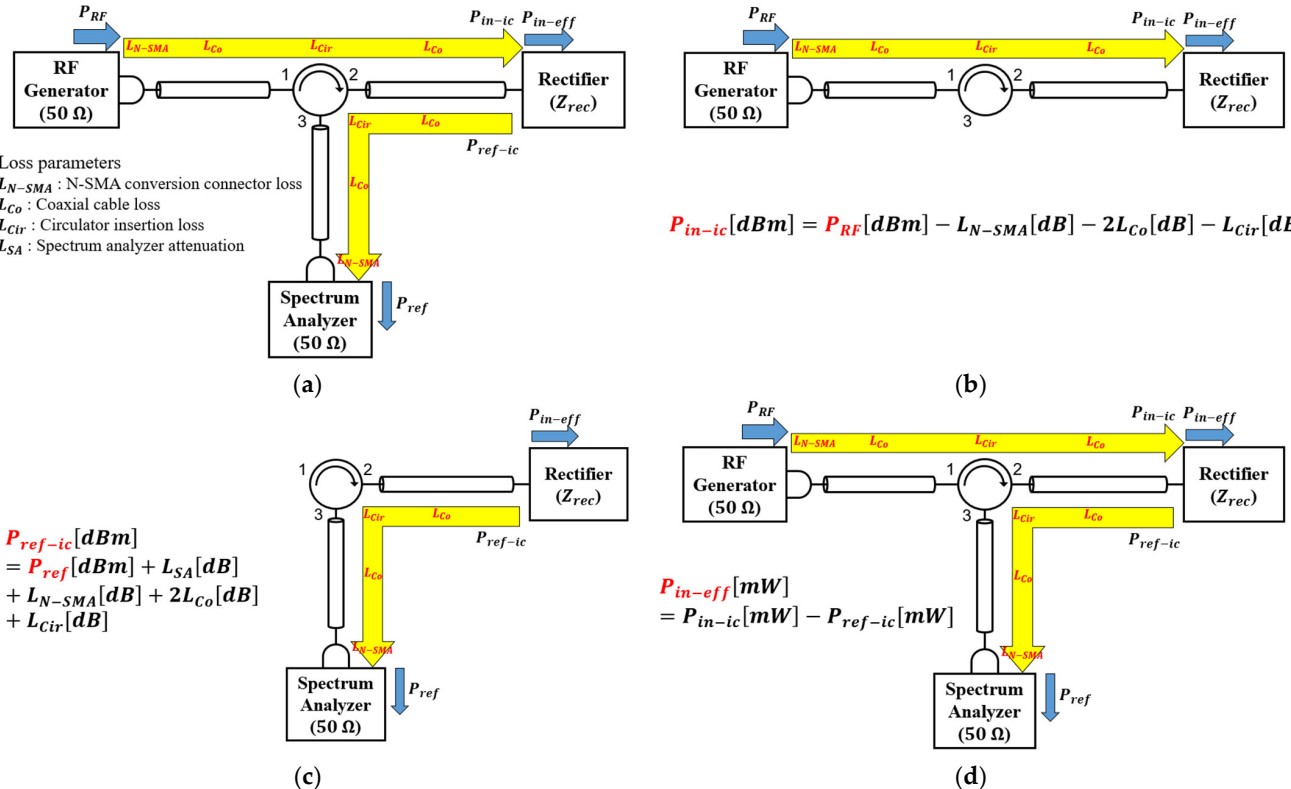

**Figure 20.** Calibration procedure of input effective power: (**a**) definition of loss parameters, (**b**) calculating the power that reaches the rectifier, (**c**) relationship between reflection power at connection point and measured power by spectrum analyzer, (**d**) equation of input effective power.

VD and SD had 1, 3, 10, 30 kΩ poly resistors connected in parallel at the output terminals. Each resistor has a transistor switch. The input power was measured with various combinations of load resistors, resulting in Figure 22. Figure 22 suggested that (1) VD has lower input power than SD does in the range of $I_{out}$ between 2 mA and 30 μA, regardless of model, SPICE and measurement, (2) $P_{in}$ of measurement is higher than that of SPICE and model in the range of $I_{out}$ between 2 mA and 30 μA and (3) the discrepancies between measurement and SPICE or model are reduced at lower $I_{out}$. Even though there are substantial differences in absolute values of $P_{in}$ between measurement and model, the proposed design methodology is valid for MWPT to supply power to low-power sensor/RF

ICs. Note that the calibration process as shown in Figure 20 may have inaccuracy, but the cause of the discrepancies in input power was not identified.

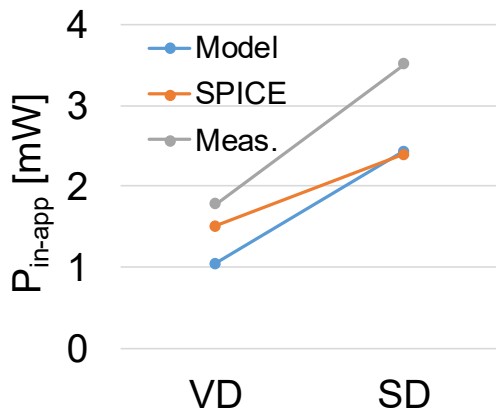

**Figure 21.** Measurement result under 50 Ω system at $P_{out}$ of 100 μW (1 V, 100 μA).

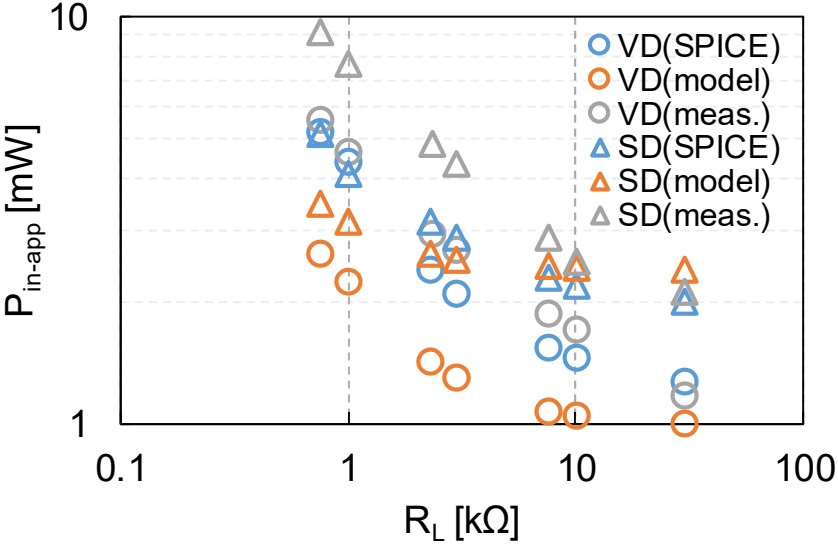

**Figure 22.** Input apparent power vs. load resistance $R_L$.

## 5. Discussion

A    Antenna impedance

In this paper, a circular loop antenna is used for one of the antenna candidates; however, the real part of the impedance drastically fluctuates by changing the loop radius near the $2\pi b/\lambda = 0.4$. Therefore, attention must be paid when using the loop antenna in actual [17].

The demonstration showed that the antenna impedance is required to be small in the real part and to be large in the imaginary part to lower the input power. Thus, it is necessary for WPT antennas with direct matching between antenna and rectifier to have high quality factors at the specific frequency. This result is matched with the previous work [18] using the high impedance antenna.

B    Relationship between rectifier input impedance and input power

In Model 2, apparent power and effective power were expressed as below.

$$P_{in-app} = \frac{|Z_{in}||Z_2|^2}{2|Z_1|^2|Z_3|^2} \tag{17}$$

$$P_{in-eff} = \frac{(R_{ant} + R_{ic})|Z_2|^2}{2|Z_1|^2|Z_3|^2} \tag{18}$$

where $Z_1$, $Z_2$ and $Z_3$ are defined in Figure 7b, and $R_{ic}$ is defined by Equation (19).

$$R_{ic} = Re(Z_1) \tag{19}$$

Equations (17) and (18) show that $P_{in-app}$ and $P_{in-eff}$ monotonically increase as $R_{ant}$, because only $Z_{in}$ includes $R_{ant}$. Except for too large $L_{bw}$ and $L_{ms}$, $P_{in-app}$ and $P_{in-efff}$ decrease as $|Z_{ic}|$ increases because the numerator has more impact on $|Z_{ic}|$ than the denominator does. Thus, $R_{ic}$ needs to be sufficiently high to reduce $P_{in}$. $R_{ic}$ mainly comes from $R_R$. Therefore, the elements connected in parallel with $R_R$ can contribute to $P_{in}$. It would be good to investigate the impact of the parasitic parameters on $R_{ic}$ and $P_{in}$. When sweeping each of the parasitic parameters, the other parameters are used as listed in Table 2. The sensitivities of $R_{ic}$ and $P_{in-app}$ on each of $L_{ms}$, $C_{ms}$, $L_{bw}$, $C_{pad}$, $R_{sub}$ and $C_{esdp}$ are as shown in Figures 23 and 24, respectively.

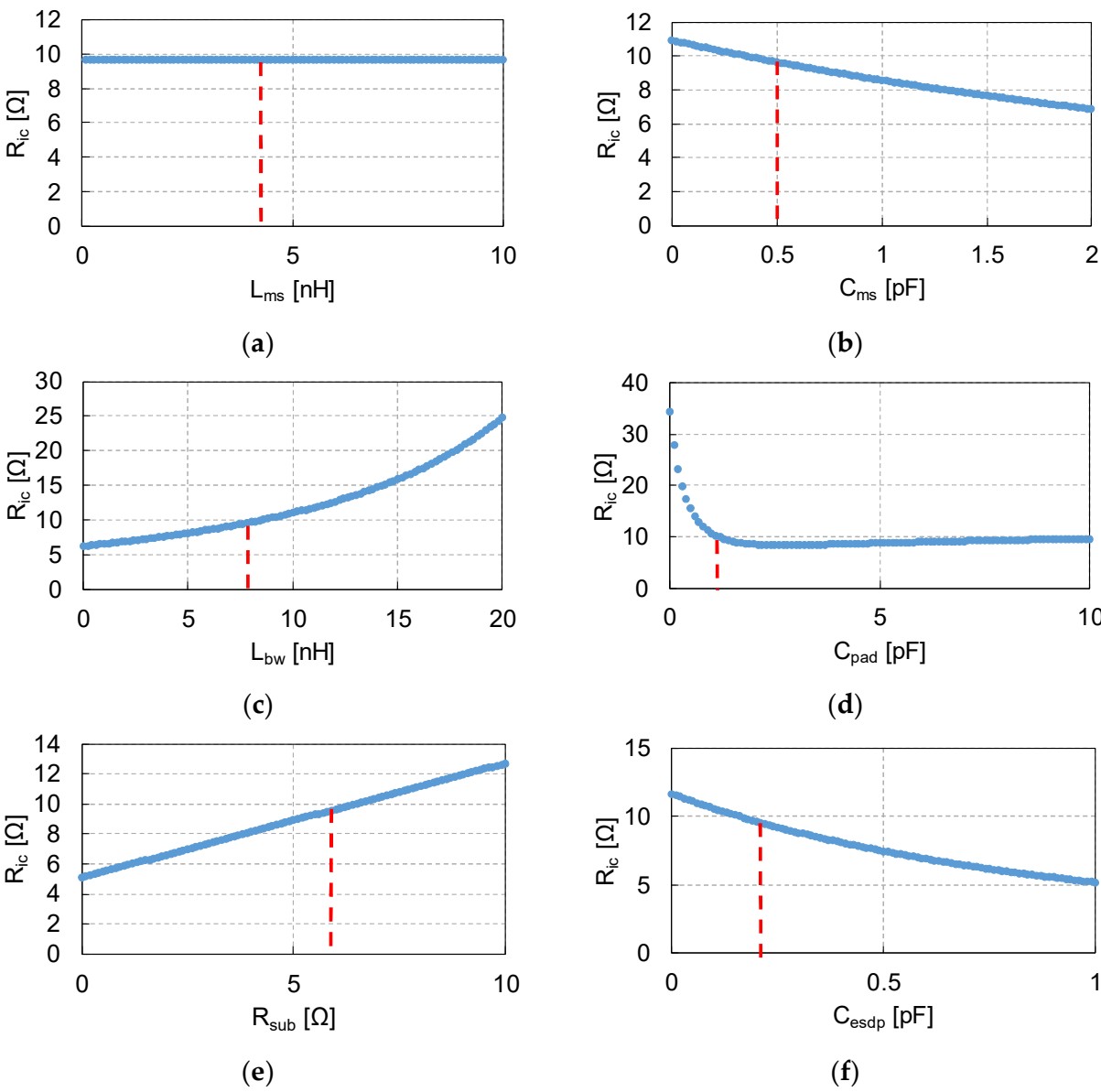

**Figure 23.** $R_{ic}$ as a function of each circuit parameter: (**a**) $L_{ms}$ sweep, (**b**) $C_{ms}$ sweep, (**c**) $L_{bw}$ sweep, (**d**) $C_{pad}$ sweep, (**e**) $R_{sub}$ sweep, (**f**) $C_{esdp}$ sweep.

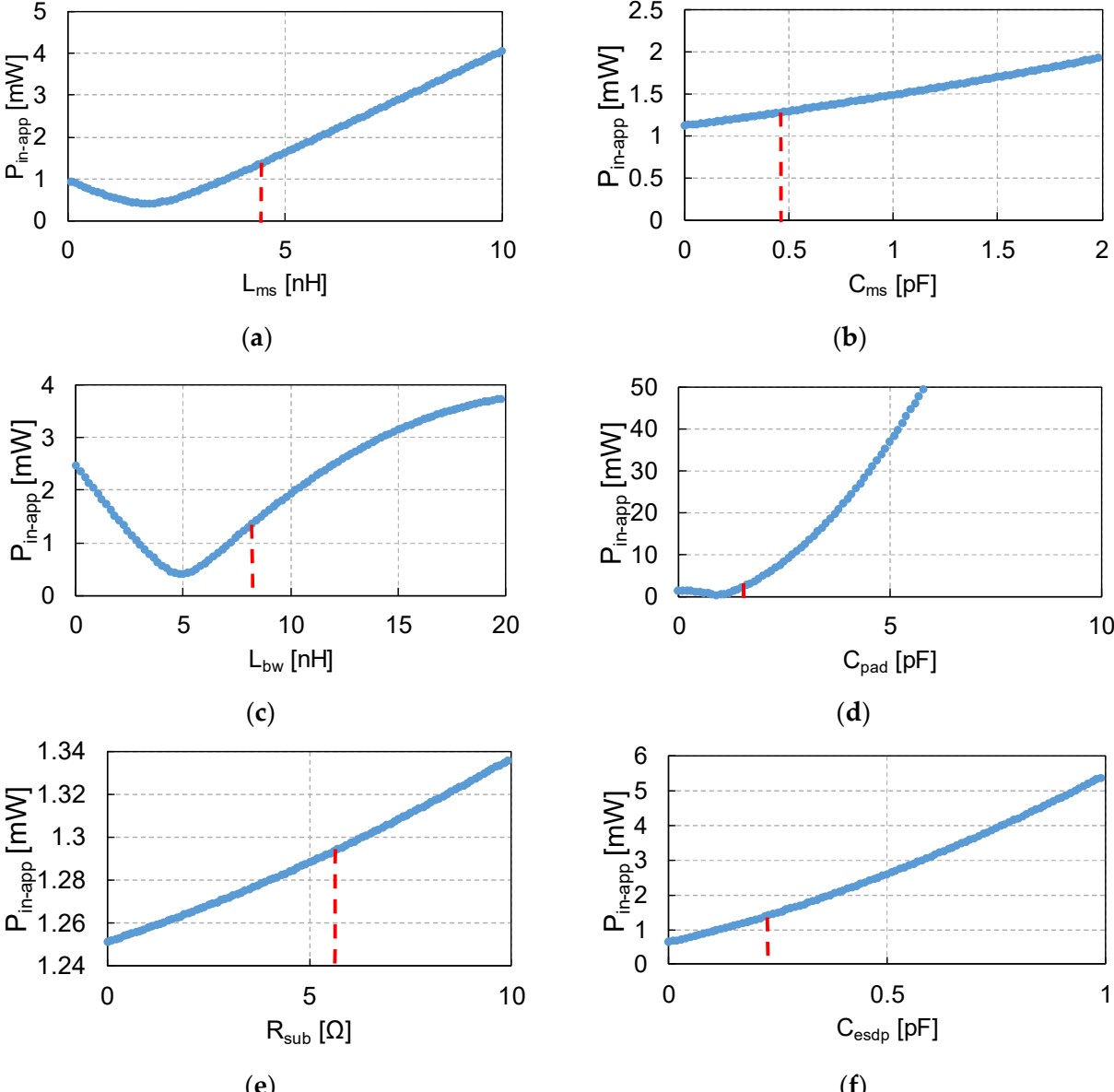

**Figure 24.** $P_{in-app}$ as a function of each circuit parameter: (**a**) $L_{ms}$ sweep, (**b**) $C_{ms}$ sweep, (**c**) $L_{bw}$ sweep, (**d**) $C_{pad}$ sweep, (**e**) $R_{sub}$ sweep, (**f**) $C_{esdp}$ sweep.

Figures 23d and 24d show that $C_{pad}$ is the largest contributor to $R_{ic}$ and $P_{in}$. A reduction in $C_{pad}$ needs to be highlighted for a high efficiency on-chip rectifier. The lower input power region of the contour plots of Figure 14 can be extended toward the higher $R_{ant}$ direction by varying each parasitic parameter in the direction of increasing $R_{ic}$. Figure 25a,b compares $P_{in}$ when $C_{pad}$ is reduced from 1.3 pF in Figure 25a to 0.3 pF in Figure 25b. The region of $P_{in-app} < 0.5$ mW can be widened in both $R_{ant}$ and $X_{ant}$ directions when $C_{pad}$ is reduced from 1.3 pF to 0.3 pF.

There are optimum values for $L_{bw}$ and $L_{ms}$ with respect to reduction in $P_{in}$, whereas lower values in $C_{ms}$ and $C_{esdp}$ can lower $P_{in}$. $P_{in}$ has no significant impact on $R_{sub}$.

In this demonstration, there were no antennas that pass through the $P_{in}^{match}$ in the considered circuit configuration. Even in such a case, the proposed exploration method to minimize input power can be used to determine the best rectenna to have low input power without running many circuit simulations.

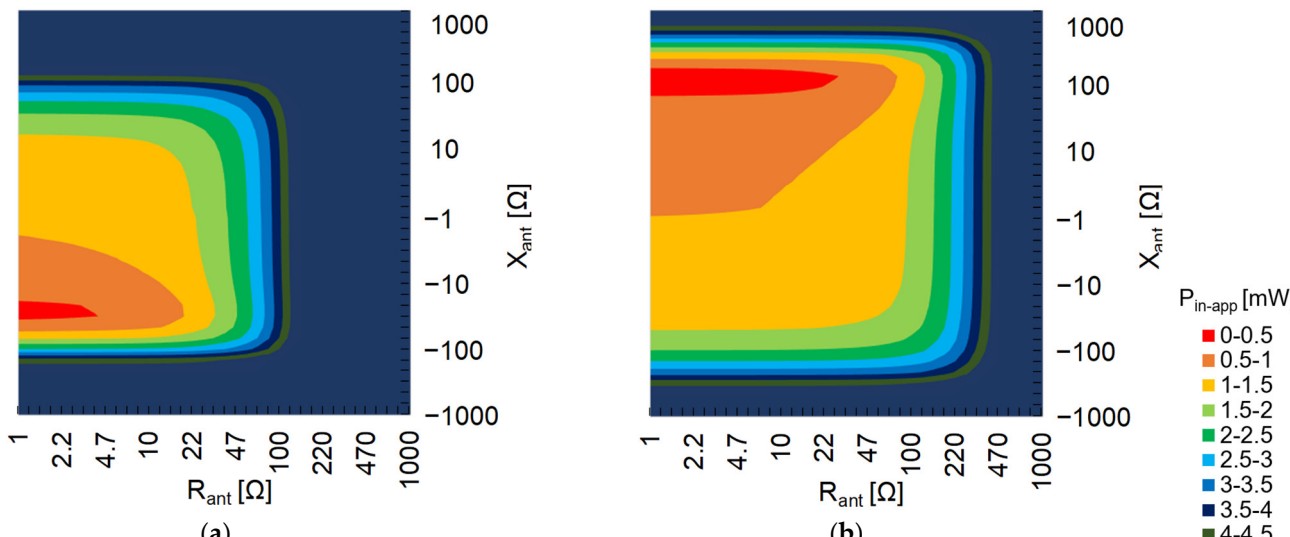

**Figure 25.** Change in input apparent power contour plots when $C_{pad}$ is varied (VD + 10 kΩ load): (**a**) $C_{pad} = 1.3$ pF, (**b**) $C_{pad} = 0.3$ pF.

C     Limitation of the proposed linearized rectenna model

Model 2 assumes that the microstrip line can be expressed by a single L-C lumped constant circuit model. When the line length is as long as the wavelength of the microwave of interest, Model 2 would need to be revised with more numbers of lumped circuits.

## 6. Conclusions

We proposed a highly efficient design flow with much less circuit simulation runs for minimizing rectenna input power in order to increase the distance from power transmitters to sensor ICs, based on the linearized rectenna model, in the case that a matching circuit is not used between antenna and rectifier, for minimum cost. The models have been validated with SPICE simulation and measured results of a single diode and voltage doubler fabricated in 65 nm CMOS. The input power to generate 100 µA at 1 Vdc was measured and compared. The model, SPICE and measurement are in good agreement with each other that VD has 30–50% lower input power than SD does. In addition, the relationship between the rectifier input impedance and the input power satisfies the target output power to investigate which parasitic parameters are most effective to minimize the input power. As a result, it was confirmed that the pad capacitance $C_{pad}$ and the parasitic capacitance of microstrip line $C_{ms}$ are most effective. By using the proposed design flow in this paper, one can determine an initial circuit composed of a rectifier topology and a target antenna impedance without repeating a large number of circuit simulations.

**Author Contributions:** Conceptualization, T.T.; methodology, T.H. and T.T.; software, T.H.; validation, T.H. and T.T.; formal analysis, T.H. and T.T.; investigation, T.H. and T.T.; writing—original draft preparation, T.H.; writing—review and editing, T.T.; funding acquisition, T.T. All authors have read and agreed to the published version of the manuscript.

**Funding:** This research received no external funding.

**Conflicts of Interest:** The authors declare no conflict of interest.

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
