# Peer review of "Design Space Exploration of Antenna Impedance and On-Chip Rectifier for Microwave Wireless Power Transfer"

_electronics, doi:10.3390/electronics11193218_

Round 1
Reviewer 1 Report
After reviewing the article entitled " Design space exploration of antenna impedance and on-chip rectifier for microwave wireless power transfer", here are my comments:
1- What is the key novelty of this article? Please highlight your contribution.
2- Please enrich the conclusion and support it with data.
Reviewer 2 Report
In this paper, the authors discuss a design methodology to efficiently determine the best
combination of rectenna to minimize the input power for MWPT. The author proposed an interesting linearized equivalent circuit model to include the microstrip line connecting the antenna and the rectifier. On this basis, the design with three steps flow is proposed. The model and the design flow are verified. Generally, the paper is well organized. However, the following issues should be clarified.
(1) In the introduction part, please summarized the main contribution of this research.
(2) It suggests other rectenna models should be compared to show the superior performance of the proposed methods.
(3) What are the limitations of the proposed linearized rectenna model?
(4) In Fig.22, why there the substantial difference in absolute values of Pin between measurement and model? Please explain.
(5) Some latest references should be added.
